# The Multifactorial Role of PARP-1 in Tumor Microenvironment

**DOI:** 10.3390/cancers12030739

**Published:** 2020-03-20

**Authors:** Juan Manuel Martí, Mónica Fernández-Cortés, Santiago Serrano-Sáenz, Esteban Zamudio-Martinez, Daniel Delgado-Bellido, Angel Garcia-Diaz, Francisco Javier Oliver

**Affiliations:** Instituto de Parasitología y Biomedicina López Neyra, CSIC and CIBERONC, Instituto de Salud Carlos III, 18016 Granada, Spain; jmmmc@ipb.csic.es (J.M.M.); monica.fernandez@ipb.csic.es (M.F.-C.); s.serrano@ipb.csic.es (S.S.-S.); ddelgado@ipb.csic.es (D.D.-B.); agdiaz@ipb.csic.es (A.G.-D.)

**Keywords:** Tumor microenvironment, PARPs, PARylation, hypoxia, autophagy, PARP inhibitors

## Abstract

Poly(ADP-ribose) polymerases (PARPs), represent a family of 17 proteins implicated in a variety of cell functions; some of them possess the enzymatic ability to synthesize and attach poly (ADP-ribose) (also known as PAR) to different protein substrates by a post-translational modification; PARPs are key components in the cellular response to stress with consequences for different physiological and pathological events, especially during neoplasia. In recent years, using PARP inhibitors as antitumor agents has raised new challenges in understanding their role in tumor biology. Notably, the function of PARPs and PAR in the dynamic of tumor microenvironment is only starting to be understood. In this review, we summarized the conclusions arising from recent studies on the interaction between PARPs, PAR and key features of tumor microenvironment such as hypoxia, autophagy, tumor initiating cells, angiogenesis and cancer-associated immune response.

## 1. Introduction

### 1.1. PARP Family of Proteins

Poly (ADP-ribose) polymerases (PARPs), more recently named ADP-ribosyl transferases (ARTs) [1] are a family of enzymes characterized by the presence of a 50 amino acid sequence called “*PARP signature*”. This allows some members of the family to synthesize and transfer ADP-ribose to a large number of substrates through a process referred as poly(ADP-ribosyl)ation or PARylation [2]. The most representative member of this family is PARP-1, firstly described by Chambon et al. in 1963 [3]. Since then, many studies have deciphered both structural and biological aspects of this family, as well as the pathological consequences of the misregulation of its members.

The best studied PARP enzyme is the original constituent of the family PARP-1. Encoded by the region 1q41-q42, this protein has a molecular weight of 114 kDa and it is constitutively expressed. In fact, PARP-1 promoter owns characteristics usually revealed in housekeeping genes. Consistently, in all tissues the mRNA of PARP-1 is present, albeit at varying levels [4]. Nevertheless, PARP-1 modulation is mainly settled at the protein level and not at the mRNA level.

The PARP superfamily contains 17 members sharing on their catalytic domain the conserved “PARP signature” (Figure 1).

As this signature is not sufficient to provide a functional classification, the categorization must be established regarding their architecture and different enzymatic functions. Although firstly classified into three subgroups [6,7], this family is now grouped into five subfamilies [8,9]:DNA-dependent PARPs. Active during DNA damage thanks to their DNA-binding domain that consist in three zinc finger and a nuclear localization signal in the case of PARP-1 (ARTD1) [10]. Other members of this group are PARP-2 (ARTD2) and PARP-3 (ARTD3).Tankyrases. Containing Ankyrin-domain repeats responsible for protein-protein interactions. Very specific of this subfamily are the sterile α motifs (SAM), also related with protein-protein interactions: Tankyrase-1 (PARP-5A, ARTD5) and Tankyrase-2 (PARP-5B, ARTD6).CCCH PARPs. Containing CCCH motifs of the CX_7–11_CX_3–9_CX_3_H type, this domain is related with RNA-binding: TIPARP (PARP-7, ARTD7), PARP-12 (ARTD12) and PARP-13 (ARTD13).Macro-PARPs. Bearing macrodomain folds. They mediate the migration of the proteins to poly (and maybe also mono) ADP-ribosylation sites: BAL1 (PARP-9, ARTD9), BAL2 (PARP-14, ARTD8) and BAL3 (PARP-15, ARTD7).Other PARPS. Proteins that do not fit into any of the previous classifications [9]; PARP-4 (ARTD4), PARP-6 (ARTD17), PARP-8 (ARTD16), PARP-10 (ARTD10), PARP-11 (ARTD11) and PARP-16 (ARTD15).

Being the most active member of this family, the PARP-1 structure was firstly described in 1984 [11] and it is composed of three domains (Figure 2):

DNA-binding domain: Involved in DNA interaction, interdomain cooperation, chromatin condensation and protein–protein binding.Automodification domain: Serve as acceptor during auto PARylation [12]. And mediates protein–protein interactions [13].NAD-binding domain: Serves as the catalytic domain, it contains the “PARP signature” sequence responsible for the PAR synthesis.

In order to perform its PARylation activity, PARP-1 and PARP-2 performs both poly(ADP-ribosyl) synthetase and transferase enzymatic activity. The PAR life cycle is described as follows [7,14] (Figure 3).

Initiation phase: Firstly, poly(ADP-ribose) synthetase activity catalyzes the breakage of the glycosidic bond between nicotinamide and ribose on the NAD+ molecule. Through this process of oxidation, ADP-ribose is formed. Subsequently, ADP-ribose is covalently attached to different acceptor proteins via formation of an ester bond between the protein (through glutamate, aspartate or lysine residues) and ADP-ribose.Elongation and branching reaction: In addition, PARP-1-mediated poly(ADP-ribosyl) transferase activity is able to catalyze the reactions responsible of elongation and branching, using more ADP-ribose units obtained from NAD+.PAR degradation: Poly(ADP-ribose) glycohydrolase (PARG) [15] mediates PAR degradation. PARG is presented in three different isoforms: PARG99 and PARG102 (of 99 and 102 kDa, respectively), both located in the cytoplasm; and PARG110 (an isoform of 111 kDa), that is located mostly in the nucleus [5].Ester bond breakage: Once PAR has been degraded, the firstly attached mono(ADP-ribosyl) moiety bond to the acceptor protein is removed by the ADP-ribosyl protein lyase [16].AMP and NAD recycling: Free poly(ADP-ribose) and ADP-ribose monomer are the final products of PAR degradation, this latter molecule can cause protein damage through glycation processes. ADP-ribose pyrophosphatase [17] converts this free ADP-ribose into AMP and ribose 5-phosphate, generating compounds much less reactive and more likely to be used in order to obtain new NAD+ [18].

As explained previously, the best described effect of PARP activation is PARylation, or covalent protein modification by PAR. This process may affect PARP itself or other proteins that become PARylated. Nevertheless, other mechanisms underlie PARP activation and PAR synthesis. They are summarized below:

Bind of different targets to free PAR in a non-covalent way [19,20]. Consistently, since the first PAR-binding domain was described [21], two new PAR-binding motifs were discovered in recent years, i.e., a PAR-binding zinc finger motif [22] and a histone macrodomain [23].Free PAR can also act as a relevant signaling molecule. The discovery of a type of cell death induced by this molecule is a clear example. In this process, to as PARthanatos, the release of apoptosis-inducing factors (AIF) from the mitochondria is triggered [24,25,26,27].PARP activation generates an important reduction of its substrate NAD+ after high DNA breakage accumulation. This depletion has important consequences on cell survival [28].

These molecular events are responsible for the different events and function alterations that take place after PARP activation (Figure 4).

The ‘‘central dogma’’ of PARylation asserts that during DNA damage PARP-1 is activated, on both single strand breaks (SSB) and double strand breaks (DSB). In fact, DNA strand breaks rises up to 500 times the basal activity of PARP-1 [29]. Consistently, DNA alkylating agents as well as reactive species of both oxygen and nitrogen (ROS and RNS, respectively) have been proved to act as triggers of PAR accumulation in different cell types [30]. However, different results have shown that PARP-1 is also stimulated in the absence of DNA damage. This has been observed in the presence of non-B DNA structures like bent, cruciform DNA or stably unpaired DNA [31].

In addition, PARP-1 activity can be modified in addition to DNA-related status; several post-translational modifications also alter PARP-1 activity. PARP-1 is implicated in multiple signaling pathways often involving a kinase phosphorylating different aminoacids of PARP-1, causing its activation or inhibition. Examples of this topic include PARP-1 being phosphorylated by ERK1/2 at serine 372 and threonine 373, deriving in PARP-1 activation [32], Also c-Met phosphorylates PARP-1, specifically at the Tyr907 causing its activation. [33,34,35]. Other phosphorylations can have the opposite effect, leading to PARP-1 inhibition, this is the case of the one performed by AMPK at Ser177 [36] and the one mediated by the protein kinase C [37]. PARP1 regulates the c-Jun N-terminal kinase (JNK) pathway, which is a driver of tumor development and treatment response. Based on that, PARP inhibition could be potentially therapeutically beneficial in ovarian cancer taken the elevated JNK activity. Furthermore, PARP1 inhibitors promote Akt [38].

Additionally, PARP-1 is regulated through other post-translational modifications different to phosphorylation but equally necessary for PARP to develop its function. PARP-1 is acetylated by p300/CBP at the lysines 498, 505, 508, 521 and 524 [39]. Also remarkable is the ubiquitination at the lysine 486 and 203 performed by the E3 ligase regulating PARP1 activity [40]. PARP-1 even undergoes ADP-ribosylation, as the described within the serine S499 [30,41].

### 1.2. Tumor Hypoxic Response and PARP-1

When an avascular tumor grows over a few cubic millimeters (depending on the tissue density and irrigation), the central parts of the mass experience low oxygen concentration and nutrients deprivation [42]. The response to this hypoxic environment has proven to be a key factor characterizing the tumor microenvironment and the disease outcome. It induces metabolic adaptations like glycolysis activation [43], cancer stem cell regulation [44], tumor exosome production [45] and the crucial tumor neovascularization [46]. New vessel formation supplies the hypoxic tumor with nutrients and oxygen, allowing the stressed cells to survive and divide, promoting mass formation and increasing cell plasticity, migration and aggressiveness [47,48,49]. The relevance of this pathway was vindicated when its discoverers were awarded the Nobel Prize of Medicine in 2019.

Furthermore, hypoxia generates resistance to different therapies. On the one hand, poorly vascularized areas make chemotherapy and immunotherapy difficult to disseminate. On the other hand, low oxygen levels make radiotherapy less effective at generating massive levels of toxic reactive oxygen species (ROS). This is the reason why hypoxic areas are known to survive treatments and are the ones to generate relapses and develop metastasis [50].

The hypoxic response is mediated by the stabilization of the hypoxia inducible factors (HIFs), a family of transcription factors composed of three alpha chains: HIF1α, HIF2α and HIF3α which are stable only during hypoxia, and one beta chain: HIF1β, which is constitutively stable. These proteins are active as heterodimers of one of the HIFα chains with the HIF1β. The heterodimers bind to the hypoxic response elements (HRE), a sequence of nucleotides located at the promoters of hypoxic inducible genes causing their overexpression. These different α/β heterodimers present similar but not identical targets and are differently expressed among tissues [51].

PARP-1 has been shown to interact with both HIF1α and HIF2α, affecting their stability and activity (Figure 5).

The crosstalk between HIF-signaling pathways and PARP-1 has been described in models of skin carcinogenesis where a reduction in HIF1α protein and mRNA is observed during PARP-1 inhibition or knockdown [52]. In the same way, in mice brains, a reduction in HIF1α accumulation during hypoxia is observed after PARP inhibition, having an impact reducing the expression of hypoxic genes like adrenomodulin or erythropoietin [53]. This dependence of HIF1α on PARP1 status can be explained by the fact that PARP-1 and HIF1α are known to form a complex during hypoxia [54]. Moreover, during hypoxia, the presence of nitrosative and oxidative stress induced via oxidative phosphorylation [55] is relevant, leading to mitochondrial inhibition and overactivation of PARP-1, promoting the stability of HIF1α [53]. It has also been described how the PARP inhibitor veliparib can sensitize both oxic and hypoxic cells in prostate and lungs to radiotherapy [56].

HIF2α is also important during tumorigenesis; tissues where HIF1α is the predominant protein become much more dependent on HIF2α during cancer progression [57]. HIF2α is related with aggressiveness and neovascularization as well [58]. Moreover, HIF2α is known to interact with PARP-1, resulting in changes in its stability and activity. HIF2α is similar to HIF1α but has different expression patterns among tissues and some of their targets do not overlap [51]. The relation between HIF2α and PARP1 has not been deeply studied but it has been proved in several cell types that PARP-1 knockdown or inhibition reduces HIF2α accumulation in the hypoxic context [59]. It has also been described that PARP-1 binds to the HIF2α promoter (but not to HIF1α promoter) controlling its transcriptional induction [60]. However, more research is needed in order to fully comprehend their interaction and its consequences.

### 1.3. Angiogenesis, Vasculogenic Mimicry and PARP-1

As previously described, blood vessels formation is a central aspect during tumor development. Vascularization not only distributes nutrients and oxygen, it also removes metabolic sub products and provides access for the immune system and different treatments.

The name ‘angiogenesis’ was used for the first time around 1900 and it was not used to refer to the tumoral context until the 1960s [61]. Since then, different varieties of angiogenesis have been depicted in cancer: Sprouting angiogenesis, micro vessel growth and microvascular proliferation [62]. Emerging studies showed the central role of this process in tumor progression [63]. Just a few tumors are able to progress without angiogenesis induction, while the vast majority of tumors present a combination of angiogenic with non-angiogenic areas [64,65]. Traditionally, treatments against angiogenesis have focused blocking new vessel growth and also trying to dismantle the existing ones, hence starving the tumors by depleting their nutrients and oxygen supply [66]. Interestingly, it has recently being reported in the PAOLA1 study (ESMO 2019) a Progression Free Survival benefit for ovarian cancer patients treated with a combination of olaparib with bevacizumab versus bevacizumab alone [67].

It is known that PARP inhibitors present antiangiogenic activity both in vitro and in vivo [52,68,69,70].

However, blood supply study in the cancer context became more complex when Maniotis et al. presented, in 1999, new findings describing cancer cells that coated vascular channels; these structures were composed of non-endothelial cells that contained erythrocytes and immune cells. This process was defined as vasculogenic mimicry (VM), and it is described as the de novo generation of a network created in a 3D matrix in vitro, composed by perfusable vasculogenic-like vessels, with properties similar to the matrix rich network described in aggressive tumors [71].

Through different studies performed on models of human uveal melanoma and cutaneous melanoma, the initial characterization of VM was performed. Since then, VM has been described in other malignancies like those of the kidney, lung, bladder, pancreas, prostate, gliomas, sarcoma, ovary, breast, head and neck cancer, and the list is still growing. In survival analyses, patients carrying VM in their tumors have been seen to present a reduced clinical outcome when compared with patients not expressing VM [72]. In 2000, a seminal study using arrays of gene expression in highly malignant melanoma cells (C8161, MUM 2B) vs poorly malignant melanoma cells (MUM 2C, C81-61) showed differences in gene expression among both groups [73]. Some of the many genes that showed significant differences were VE-cad. The depletion of VE-cadherin with specific siRNA and VE-Cad antibody abolished VM in a 3D in vitro model [74]. VE-cadherin is a trans-membrane protein usually located in the endothelium, necessary for the endothelial stability through its activity in cell–cell adhesion (Figure 6).

This protein presents the activities described for classic cadherins. Since the description of VM, VE-cadherin cannot be considered a marker of exclusively endothelial cells. VE-cadherin expression has been associated with aggressiveness on different tumors carrying VM. In this direction VE-cadherin can be found in highly aggressive VM tumor cells but not in poorly aggressive ones. Moreover, its down regulation in melanoma implied the loss of capacity to form VM [74].

It has been shown that activation through VEGFR-1 or through its co-receptor NRP-1 contribute to VM [75,76] while the potent inhibitor of αν integrins cilengitide displays anti-melanoma activity through the inhibition of VM [75]. A contrary role of VEGF signaling in VM has been also proposed. While is commonly accepted that VEGF and receptors like VEGFR1 or VEGFR co-receptor (NRP-1) promote VM, it has been also proposed that VM could also appear in contexts lacking this pathway: In this situation VEGF would promote angiogenesis, and its inhibition would potentiate other survival strategies including VM [77]. A study in 2013 by Rodriguez et al. gave a closer view into VM complexity, studying the interplay between PARP inhibition, epithelial-mesenchymal transition and VM. The use of the PARP inhibitors PJ-34 and KU0058948 in a murine melanoma cell line decreased the expression of VE-cadherin, as well as its phosphorylation in the residue of tyrosine 658. This resulted in the depletion of VM formation in in vitro assays [78].

The use of angiogenesis inhibition in combination with other treatments has been approved by the FDA in some scenarios; this is the case of an anti-VEGF specific antibody (bevacizumab, Avastin, Roche) used in combination with chemotherapy or different cytokines therapies for late-stage advanced metastatic cancers (including renal cell cancer, colorectal cancer, non-squamous non-small cell lung cancer and breast cancer) [79]. On the other hand, angiogenesis inhibitors in combination with immune checkpoint in the treatment of breast cancer showed no clear benefits and are still under evaluation, so there are no current FDA-approved indications for their use in breast cancer [80]. Current studies show how the combination of angiogenesis and PARP inhibitors will be likely safe due to non-overlapping toxicities, and it might be expected that PARP inhibitors could be used in this context at full mono-therapy dosages.

## 2. Immuno-Response Modulation by PARP

There is increasing knowledge describing the immunological role of the PARP family that supports the combinatorial uses of PARP inhibitors (PARPi) and immunotherapies against cancer. It is known that PARPs have a role during inflammation, innate immunity and in immune cells. PARP enzymes interact with transcription and adhesion factors which are involved in the regulation of cytokines and inflammatory mediators related to different aspects of inflammation [81]. Several studies showed a PARP-1-dependent activation of NFĸB, a major transcription factor during the inflammation process [82,83,84]. PARP-1 is required to trigger NEMO SUMOylation and monoubiquination which is necessary for NEMO and NFĸB activation [84]. There are a complete set of other transcription factors involved in inflammation which are also modulated by PARPs like sp1 [85], NFAT [86] or SIRT1 [87] among others.

Otherwise, PARP-1 and PARP-2 regulate several common inflammatory factors and cytokines including Tumor Necrosis Factor-α, inducible Nitric Oxide Synthase or Interleukin1-β suggesting an overlapped mechanisms or regulation [88]. In addition, PARP enzymes are involved in regulating the expression of cytokines and chemo attractants like IL-6, IL-12 or CCL2. PARP-14 enhances the STAT6 regulation of the expression in Th2 cells through IL-4 induction, which is important for the immune function in the lung [89].

PARP-1 is involved in development and activation of different immune cell types like macrophages, microglia or dendritic cells. Moreover, PARP-1 and PARP-2 induce pro-inflammatory effects not only restricted to innate immune system cells but also important in dendritic cells and fibroblasts [90]. PARP enzymes are also involved in T-cells development. PARP-1 and PARP-2 expression is especially high in the lymphocyte-proliferative areas of the thymus. Moreover, inactivation of PARP-2 decreases the size of the thymus while reducing the number of CD4+/CD8+ (double positive) thymocytes, due to an affected survival of double positives [91]. In addition, other studies showed an impaired capacity to differentiate into Th2 cells in PARP-1 KO cells [92].

The stimulator of the interferon genes (STING) pathway was primarily described as a mechanism activated in response to microbial infections and DNA viruses, but also has relevance under certain autoimmune and inflammatory conditions. There is currently a list of new evidence suggesting a role of the STING pathway in tumor detection [93,94]. Activation of STING is produced by the accumulation of cytosolic DNA fragments which interact with the cGAMP synthase (cGAS), catalyzing the formation of the second messenger GAMP to activate STING [95]. After its induction, STING activates TBK1 which, in turn, phosphorylates STING and the interferon regulatory factor 3 (IRF3). Then IRF3 migrates to the nucleus, causing the overexpression of type I interferon genes, including interferon beta (IFNβ). STING and type I interferon beta signaling pathways are involved in T-cell priming and activation against tumor-associated antigens in the tumor microenvironment [96,97].

After the detection of cytosolic DNA and the activation of the cGAS-STING-TBK1-IRF3 axis leading to the activation of type I interferons, there is an observed induction of cytokines involved in T cell chemotaxis, as CCL5 or CXCL10. It has been reported that type I IFN production and CCL5 or CXCL10 expression correlate with the infiltration of cytotoxic lymphocytes CD8^+^ in several cancers [98]. Jianfeng Shen et al. reported a mechanism describing PARPi modulation of immune responses against cancer, even independently of *BRCA1/2* mutational background, through the STING pathway, further enhanced by blocking different immune checkpoints [99].

It is also relevant that tumor mutational burden (TMB) serves as a good predictor of response to immunotherapy, since it correlates with the sensitivity of tumors to the immune checkpoint blockade on immunotherapies like antiPD-1/PD-L1 or CTLA4 [100]. In other words, the immunogenicity of a given tumor depends, in part, on the mutational load and subsequently on its neoantigen repertoire. Recognition of such neoantigens is considered a major factor in the efficacy of clinical immunotherapies [101]. Nonetheless, cancers presenting elevated copy numbers such as ovarian cancer and small cell lung cancer are not immunologically hot but have extraordinary levels of damaged DNA/chromosomes and there exists controversy about the neoantigen hypothesis. Some groups have shown that the vast majority of mutations within expressed genes in cancers do not lead to the formation of neoantigens that are recognised by T cells [102,103].

Mutational load range varies over several orders of magnitude between different types of tumors [104,105], but ultimately, it will be the consequence of a balance between different factors including DNA damage and DNA repair function. Conditions affecting these factors are therapeutically exploited with different approaches, using DNA chemotherapy damaging agents or radiation. This is especially important with genetic backgrounds comprising an impaired DNA repairing machinery or inhibitors of DNA repair proteins involved in DNA damage response pathways as single agents or in combination with the DNA-damaging agents [106].

As previously indicated, PARP-2, PARP3, and especially PARP-1 became catalytically active in response to DSB, recruiting proteins that are involved in chromatin remodeling and DNA repair [107]. If the tumor immune response is modulated by the mutational load of the cells and the subsequent presence of neoantigens, it seems reasonable to hypothesize that PARPi could increase the tumor mutational burden due to an impaired DNA repair pathway function, contributing to the immunotherapy efficacy. There are currently several clinical trials exploring this possibility mainly but not exclusively in ovarian cancer [106,108].

Programmed cell death-1 (PD-1) is an immune receptor mainly expressed on activated CD4+ and CD8+ T cells or peripheral B cells [109]. Interaction of PD-1 and its ligand PD-L1 is critical to control the immune response, providing its binding constitutes an immune inhibiting checkpoint which leads to immune evasion. PD-L1 can be induced in cancer cells by a variety of stimuli such as T cell interferon gamma production or ionizing radiation (IR) [110] among others. The radiation-dependent PD-L1 activation seems to be related to DSBs and DNA repair response pathways and synergistically enhances antitumor immunity if applied together with immune checkpoint inhibitors [111]. This enhanced antitumor immunity seems to be related to increased mutational burden which increases neoantigen repertoire and tumor infiltrating lymphocytes (TILs) [112].

Interestingly, several studies also reported increased intratumoral CD8^+^ T cell infiltration and interferon production after PARP inhibition [99,113]. Nevertheless, this increased presence of antitumor immunity and TILs can be counterbalanced by PARPi-induced expression of PD-L1 which subsequently activates the PD-1/PD-L1 immune checkpoint pathway [114].

Taken together, these observations provide further rationale for the combinatorial uses of DNA damaging agents, PARP inhibitors, and immune checkpoint blockade, in order to increase the benefits of the enhanced antitumor immunity avoiding the immunosuppressive effects of PD-L1 overexpression (Figure 7).

## 3. PARylation in Autophagy

One of the main characteristics of the metabolically hyperactive cancer cells is their high demand for nutrients and oxygen from their microenvironment. This makes them vulnerable to the deficiency of both oxygen (hypoxia) and nutrients (starvation). Both situations can induce metabolic stress leading to PARP-1 over activation; during these situations, cells must activate the autophagic pathway as a response to adapt and survive. In this context it has been shown that one of the first consequences of autophagy is the activation of poly-(ADP-ribosyl)ation. The combined effect of PARP-1 activity leading to PAR modification of AMPK (the main cell energy sensor) and the nutritional status (measured via mTORC1 activation) are fundamental during the first stages of autophagy [115].

The autophagyc pathway is present in all eukaryotic cells. It consists on a “self-eating” process necessary for the maintenance of the cell homeostasis. Through this lysosomal-dependent pathway different cellular organelles, portions of the cytosol and chaperone-associated cargoes are enfolded in double-membrane spheroids called autophagosomes, being then hydrolyzed by lysosomal enzymes [116]. Different cellular stresses can fire the autophagic response: hypoxia, pH variation, mitochondrial ROS production, DNA damage, intracellular pathogens, unfolded proteins or endoplasmic reticulum (ER) stress among others.

Expanding evidence indicates how autophagy is induced after DNA breakage. Ataxia Telangiectaxia Mutated (ATM) has been described as an important link between DNA Damage Response (DDR) and the induction of autophagy [117]. In response to DNA damage by mitochondrial ROS, external toxins or irradiation, ATM is auto phosphorylated within a Mre11, Rad50 and Nbs1 (MRN) multiprotein complex that binds DSBs. Once active ATM induces the activity of AMPk and its target tuberous sclerosis protein (TSC2), this leads to the inhibition of mTORC1, promoting the formation of autophagosomes dependent of Unc-51 like autophagy activating kinase (ULK1) [118]. This is how autophagy acts as a catabolic pathway of rapid energy recovery that the cell will use to grow and expand. However, this rapid acquisition of energy has two sides; on the one hand, it responds to a pro-survival role as long as the levels of autophagy remains biologically sustainable, this way, the degradation of excess vital components of the cell is avoided; but autophagy might also become a process of cell death due to the accumulation of autophagosomes degrading essential cellular structure;, this process is called autophagic cell death (ACD).

As we know, PARPs are important guardians of the genome integrity and they have been shown to link DNA damage response with the activation of autophagy [115,119]. Autophagy is also initiated in several settings of response to chemotherapy and mediated by PARP-1. Doxorubicin treatment leads to over-activation of PARP-1, followed by ATP and NAD+ depletion, triggering the non-toxic accumulation of autophagosomes [19]. In response to methylnitronitrosoguanidine (MNNG) double knockout Bax−/−Bak−/−MEFs activate PARP-1, reducing intracellular ATP levels and activating AMPk pathway and mTORC1 down-regulation. Suppression of AMPk pathway blocks MNNG-induced autophagy and enhances cell death [19]. The same results were found in a nasopharyngeal carcinoma model exploring over-activation of PARP-1, upon ionizing radiation there is a PARylation-dependent energy depletion and up-regulation of AMPk and ULK1 pathways [120]. All these studies have demonstrated that autophagy should be contemplated as a target in cancer during the induction of DNA damage and consequently, new strategies based on the synthetic lethality concept during PARPi must be explored.

In addition to DNA damage, nutrient deprivation can be considered as the most physiological stimulus to induce reversible autophagy. Recent studies of our group have implicated PARP-1 in autophagy induced by nutrient starvation [115,121]. The absence of PARP-1 or the use of PARP inhibitors (PJ34, DPQ and Olaparib) delay starvation-induced autophagy. Mitochondrial ROS accumulation derived from starvation promoted DNA damage and PARylation signaling from PARP-1 activation, which triggers ATP depletion, AMPk activation and mTORC1 inhibition. The lack of PARP-1 activity compensates ATP depletion compromising the cascade AMPk/mTORC1 inhibition/ autophagy. Starved PARP-1-deficient/inhibited cells showed increased apoptotic cell death [115]. In order to analyze the in vivo consequences of PARP-1 ablation on autophagy both PARP-1+/+ and PARP-1−/−pups were starved for short periods of time, concluding that PARP-1−/−neonates display a deficient liver autophagy response following acute starvation, showing that PARP-1 activity and PAR formation are key players in the decision of the cell to engage autophagy [115].

The absence of PARP-1 compromises the activation of AMPk, more precisely the isoform AMPkα, reducing phosphorylation on Thr172 [121,122]. A nuclear subpopulation of AMPkα was detected in the breast cancer cell line MCF7, where it forms a stable complex with PARP-1; moreover the activation of nuclear AMPkα requires two essential events: Firstly, starvation-induced ROS must be imported to the nucleus to generate DNA damage and PARP-1 activation. Secondly, PARP-1 modifies by PARylation the nuclear population of AMPkα. Energy depletion associated with starvation was increased during non-lethal over-activation of PARP-1 in response to ROS, resulting in a feedback loop that favored the interaction between PARP-1/AMPkα and promoted the activation of LKB1. The mechanism of modulation by PARylation of LKB1 is not clearly understood but using PARPi or during specific assays with siRNA PARP-1, LKB1 activation is compromised [121].

In non-starved cells, PARP-1 forms a complex with AMPK; after starvation PARP-1 is activated and PARylation disrupt the PARP-1/AMPK complex, this triggers the nuclear export of PARylated AMPkα to the cytoplasm. This significant PARylated AMPkα population, is recognized by the active form of LKB1 promoting the phosphorylation on regulator site Thr172 of AMPkα. Again, a positive feedback loop takes place between the cytosolic energy depletion and PARylated AMPkα to potentiate the activation of the cytosolic AMPkα population by LKB1. The presence of covalently PARylated LKB1 has not been demonstrated but the effect of PARPi using PJ34 or olaparib suggested the implication of PAR on LKB1. These data place the nucleus (a classical cellular component not associated with autophagy) and the interaction PARP-1 and AMPk as an initial sensor of the metabolic alterations derived from perturbations in the nutritional extracellular status, not necessarily related with the alterations in genomic integrity. PARylation could be considered as an alert indicator of changes in the energy balance in cancer cells (Figure 8).

## 4. Cancer Initiating Cells

Cancer Stem Cells (CSCs) are defined as the tumor cell subpopulation that present remarkably high plasticity, considering “plasticity” as the ability of pluripotent cells to induce trans-differentiation. The first evidence of cancer initiating stem-like cells was described in 1994 by Lapidot et al. and Caceres-Cortes et al. after injecting different leukemia cell subpopulations in mice [123]. Through the years, CSCs were determined in a wide multitude of cancers like breast [124], gliomas [125], prostate [126], melanoma [127], lung [128,129], colon [130,131], pancreas [132], head and neck squamous cell [133], liver [134] and renal carcinoma [135].

CSCs are a very aggressive subpopulation of cancer cells characterized by their self-renewal capacity, tumor initiation ability, resistance to chemo- and radiotherapy and multi-lineage differentiation (stemness) [125,136,137]. The presence of this subpopulation in tumors is considered to be a marker of bad prognosis and therefore they have been proposed as target in cancer therapy.

For several years, it has been known that reprogramming processes are led by specific genetic programs. Yamanaka et al. demonstrated that the ectopic expression of four genes (Oct4, Sox2, c-Myc and KLF4) in mouse fibroblast transforms them in embryogenic stem cells. These cells, denominated by the authors as induced-pluripotent stem cells (iPSCs), were able to generate cells from any germ layer [138]. This breakthrough made Shiya Yamanaka win the Noble Price of Medicine in 2012. Today, these genes are known as Yamanaka factors and several researches link the expression of these genes with cancer [138,139,140]. In fact, the risk of tumor development is the most important limitation in the application of iPS cell-based therapy. Tumor cells present processes of trans-differentiation and dedifferentiation or reprogramming acquiring features of CSCs [138].

In 2012, it was demonstrated that PARP-1 is involved in the reprograming process, promoting iPSCs. An intense PARylation was detected in iPSCs. Moreover, PARP-1 knockdown reduced the capacity of iPSCs generation after Yamanaka factors overexpression. It has been found that c-Myc directly regulates PARP-1 expression and PARylation. Over-expression of PARP-1 compensates the knockdown of c-Myc in reprogramming in MEFs [141]. In 2009, Gao et al. demonstrated that PARP-1 PARylates and controls Sox2 levels, regulating its activity. They found that PARylated Sox2 increases its stability and therefore induces the expression of its target gene FGF4. PARP inhibition has been observed reverting this scenario [142,143]. To reprogram cells to iPSCs by Yamanaka factors, an epigenetic remodeling is also necessary. PARP-1 regulates modification in histones that alter the chromatin pattern driving to pluripotent cell phenotype. In this line, PARP-1 is necessary to promote the access to chromatin of Oct4 [144].

Telomerase (TERT) is a reverse transcriptase necessary for telomeres elongation in embryonic stem cells. This retrotranscriptase is highly expressed in tumors and the expression is related with stemness capacity. KLF4 directly interacts with the promoter of TERT. PARP-1 interacts with KLF4 and mediates the expression of TERT in CSCs. In fact, PARP-1 suppression dramatically reduces the recruitment of KLF4 to the promoter of TERT, reducing its expression [145]. C-Myc has also been related with TERT expression, suggesting another possible relationship between PARP-1 and telomerase [146].

Due to the high resistance to radio and chemotherapy presented by CSCs, many studies focused their interest in this subpopulation as a target for future treatment. In glioblastoma (GBM), one of the most aggressive malignancies, the presence of CSCs was proposed as the main cause of tumor relapse [138]. Furthermore, reprogramming events driving the formation of endothelial cells from tumor cells have been described [147]. The importance of CSCs in this kind of tumors led Vescovi el al. to propose the differentiation of CSCs as a target, with the aim of reducing the malignancy of this brain cancer [148]. They found that glioma stem cells treated with bone morphogenetic proteins (BMPs), cytokines belonging to TGF-β superfamily, differentiated CSCs to non-stem glioma cells. BMPs interact with membrane receptors inducing differentiation through SMADs signaling pathway. It has been demonstrated that PARP-1 negatively regulates this pathway at different levels. Ectopic expression of PARP-1 suppresses the signaling mediated by BMP. On the contrary, knockdown of PARP-1 promotes BMP-mediated differentiation. Furthermore, PARG, an enzyme responsible for PAR degradation, plays a positive role in the pathway [149,150]. On the other hand, SMAD signaling activation can be a double-edged sword in GBM because the stimulation of this pathway with TGF-β induces the expression of leukemia inhibitory factor (LIF), a cytokine that induces maintenance of “stemness” capacity [151].

## 5. Conclusions and Perspectives

The PARP superfamily consist of a group of proteins characterized by the presence of what is called a “PARP signature” on their sequences. Their main characteristic activity is the process referred to as PARylation. Through NAD+ and ATP consumption, ADP-Ribose is generated and then transferred as poly or mono ADP-ribose to different target factors. These modifications alter their activity or stability having relevant implications on the cellular metabolism. Moreover, excessive PARylation has an impact itself: free PAR can function as a signaling molecule and its synthesis may blunt NAD+ and ATP levels producing cell death.

PAR synthesis is activated mostly during DNA damage, altered DNA configuration or when PARP present some posttranscriptional modifications. As we summarized in this review, PARP activation has key repercussions on the cell fate affecting processes like DNA repair, transcriptional regulation, DNA remodeling, hypoxic response, epithelial mesenchymal transition, angiogenesis, autophagy, inflammation and cancer stem cell programming. All these processes lead to changes in survival, proliferation, differentiation, or even malignant transformation. Considering the relevance of the previously enumerated processes, it is easy to understand the relevance of PARP during cancer development. Cells overexpressing PARP will be more likely to repair DNA damage induced by genotoxic agents, they will adapt better to hypoxia and will be prone to produce metastasis through angiogenesis and EMT. Four PARP inhibitors have been already approved by the FDA (olaparib, rucaparib, niraparib and talazoparib) and they are used today as a result of their ability to generate “synthetic lethality” on BRCA 1/2 mutated tumors (see [152] for a review). However, ongoing advanced clinical trials will most likely expand their prescription as is the case for the combination of PARP inhibitors with classical therapies (Table 1) and with anti-angiogenic treatment (Table 2).

Currently, one of the limitations facing this therapeutic option is considering mostly on the BRCA mutated cells as HR deficient. Knowing that more than 100 genes are involved on this DNA repair pathway, it looks likely that BRCA proficient cells could be still HR defective due to other mutations. Measuring the whole genomic instability within a tumor by surveying the loss of genetic heterozygosity, telomeric allelic imbalance or the extent of somatic mutations, could be a more precise approach to define PARP inhibition sensibility, making more patients candidate of this therapeutic approach.

Another important limitation that needs to be worked out is the emergence of resistances, especially during PARP inhibition in monotherapy (Figure 9).

The generation of a selective force (like the caused by a drug treatment) in a highly mutagenic context can lead to the selection of resistant clones. They will then reproduce, forming a relapsed tumor resistant to the original treatment. To avoid this undesired consequence, the selective pressure can be reduced by spacing the treatments or by combining it with inhibitors for different targets, generating a constant pressure over the tumor while generating reduced selective forces.

Another way to avoid resistance is to study the genetic background and predict the more probable resistance mechanisms to arise, then combining the PARP inhibitor with secondary inhibitors for the factors related with the expected resistance. It is known that HR restoration, checkpoints activation, stalled DNA protection or drug efflux among others, can lead to PARP inhibition resistance. Combining PARP blockade with inhibitors for the possible ways of scape the treatment is being tested in clinical trials in order to make PARP inhibition more effective.

Based also in the studies summarized here, we propose that PARPi is an expansive field that may have therapeutic value beyond synthetic lethality. To this end, a precise comprehension of the implications of the different PARPs with PARylation in the complex tumor ecology is needed, including analysis of PARylation in both tumor and associated non-tumor cells, using single cell analysis and its consequences in tumor adaptation to hostile conditions. In this context, we can affirm that the use of PARP inhibitors against cancer treatment is not just a promising field but a reality and the challenge exists to widen their use by identifying new properties and deepening the role of PARP in tumor biology.

## Figures and Tables

**Figure 1 cancers-12-00739-f001:**
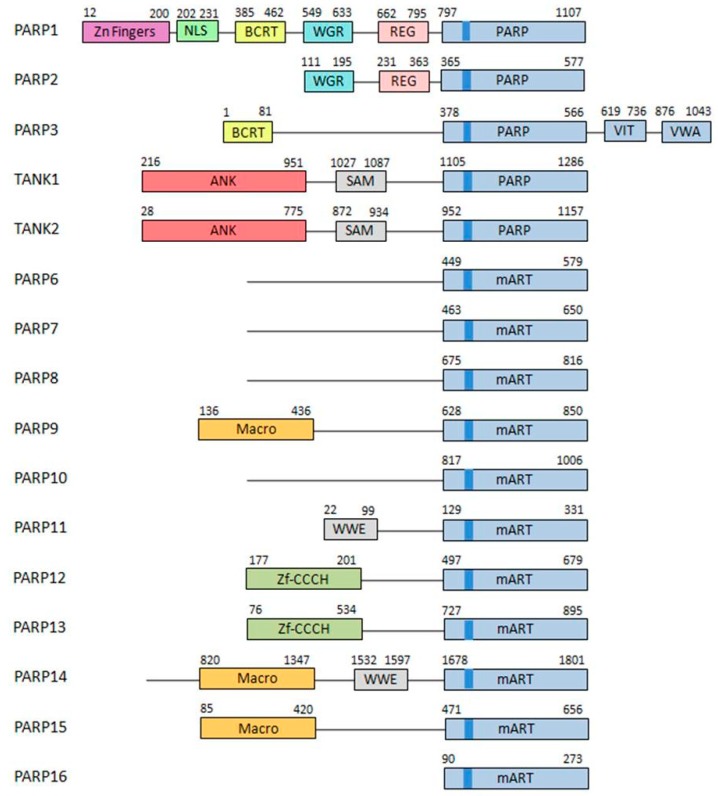
Poly(ADP-ribose) polymerases (PARP) family. The structure of the different members of the PARP family is described. Different domains are detailed in different colors. Brighter blue shows the PARP signature sequence, common throughout all the members of the family [5].

**Figure 2 cancers-12-00739-f002:**
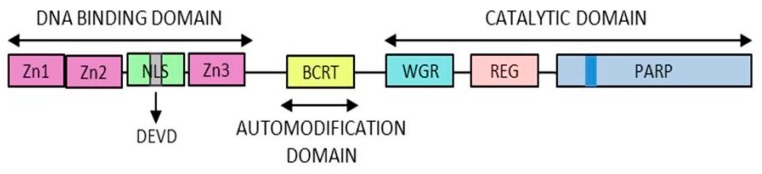
PARP1 structure. The different domains of PARP1 are presented here. The zinc fingers responsible for the DNA binding capacity, the automodification domain thanks to which PARP1 is modified by polymer and the catalytic domain containing the “PARP signature”.

**Figure 3 cancers-12-00739-f003:**
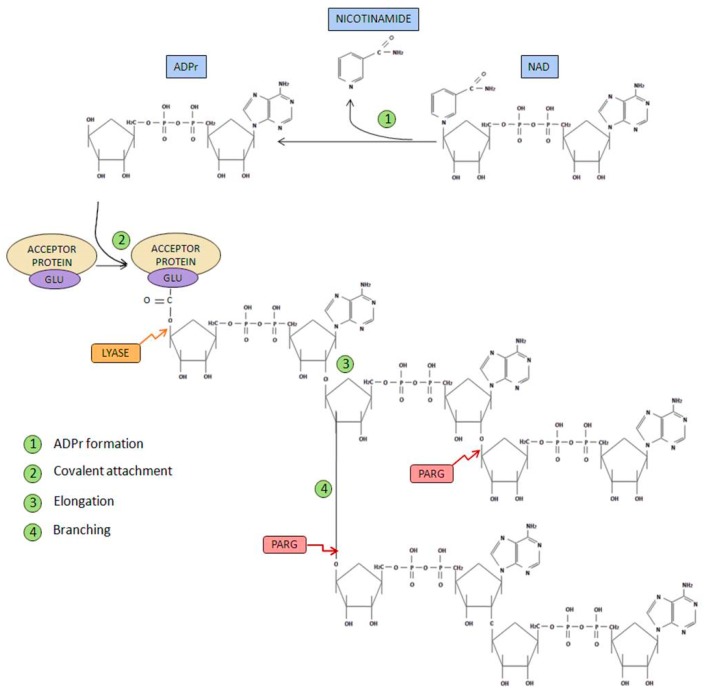
PARP metabolism. Different steps in polymer formation are shown in green. In contrast, the proteins involved in polymer degradation are shown in red and yellow [14,15].

**Figure 4 cancers-12-00739-f004:**
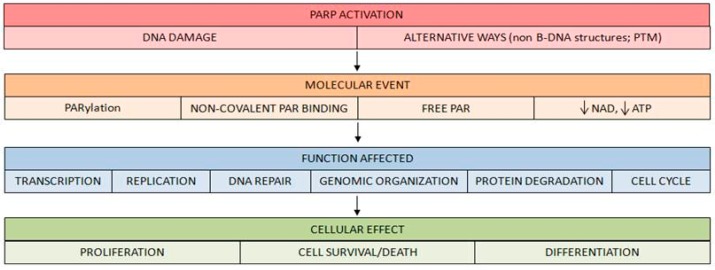
Molecular events following PARP activation. Once PARP is activated, downstream events of PARP signaling take place, involving either covalent PARylation of substrates, non-covalent binding of PAR polymer to proteins bearing a PAR-binding motif, release of free PAR to the cell or lowering cellular NAD+/ATP levels. Via these pathways, PARP/PARylation regulate functions such as transcription, replication, DNA repair, protein degradation and cell cycle, mediating various cellular phenomena such as proliferation, cell survival and cell death or differentiation.

**Figure 5 cancers-12-00739-f005:**
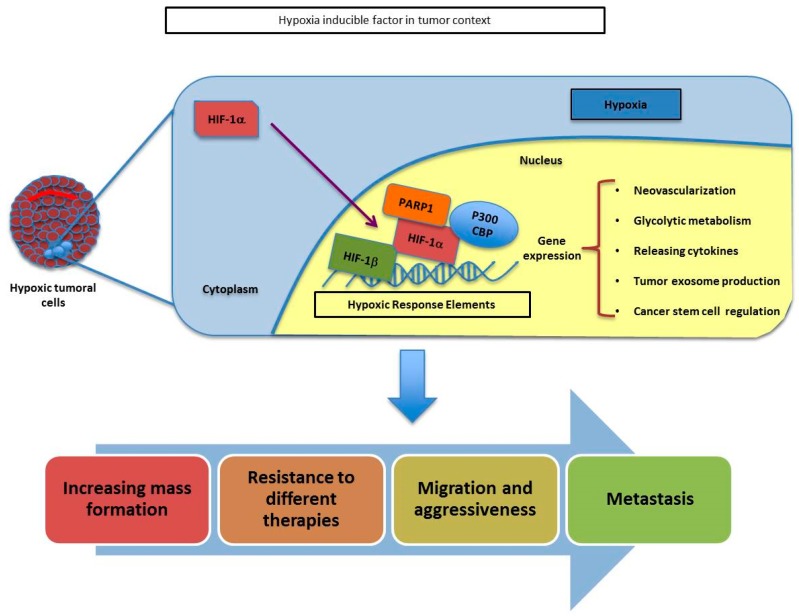
Schematic representation of HIF-α and PARP1 association throughout the adaptation of tumoral cells during hypoxia. The union of PARP1 to HIF-a helps to its stabilization and allows the transcription of several genes involved in tumor growth, resistance, migration and metastasis.

**Figure 6 cancers-12-00739-f006:**
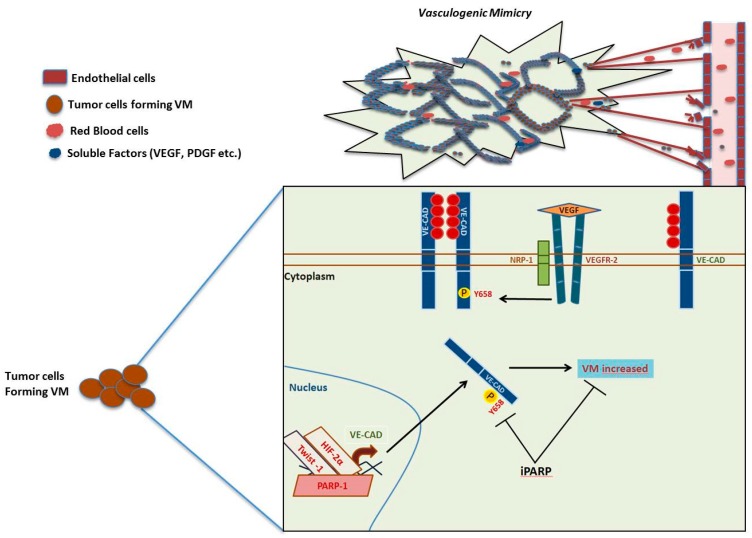
Vasculogenic mimicry pathway and PARP. VE-cadherin can be phosphorylated by stimuli of VEGF by VEGFR2 activity and the co-receptor (NRP-1). The soluble factor (VEGF) increases the phosphorylation of Y658 of VE-cadherin and the consequent internalization. Inhibition of PARP decreases the phosphorylation Y658 of VE-cadherin and finally the possibility to form VM in aggressive melanoma cells (B16F10 cells).

**Figure 7 cancers-12-00739-f007:**
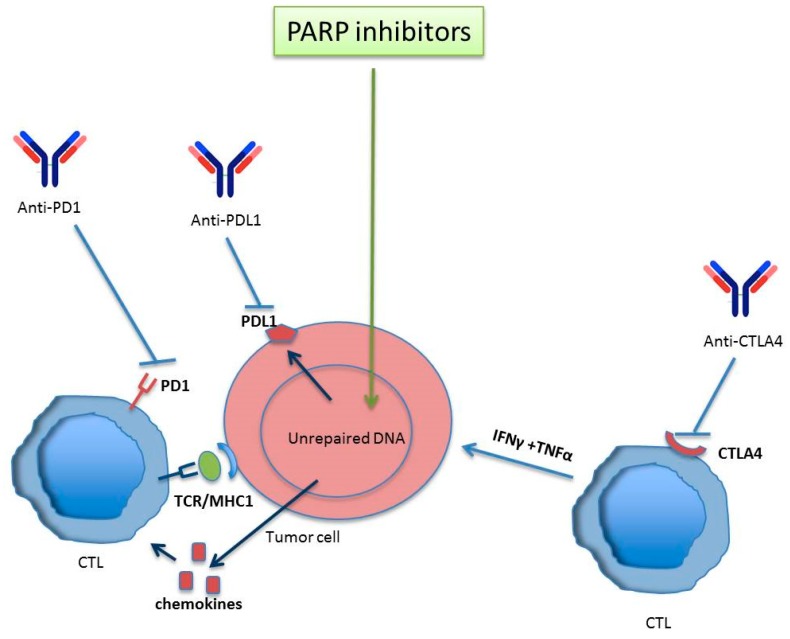
PARP inhibitors together with immune checkpoint inhibitors potentiate antitumor immune-mediated response. Activated T lymphocytes react against tumor antigens. The inhibition of PARP increases the number of lymphocytes infiltrating the tumor after the upregulation of chemokines, promoting an immune response mediated by CTLs. In spite PARP inhibitors modulate positively the upregulation of PDL-1 (favouring tumor scape from immune control) the anti-CTLA4 activates T cells to promote an antitumor response. Anti PD1/PDL-1 reverses CTL inhibition provoked by PARP inhibitor –induced PDL-1 expression. In this way, anti-PD1/PDL-1 can synergize with PARP inhibitors to ammeliorate antitumor immune response.

**Figure 8 cancers-12-00739-f008:**
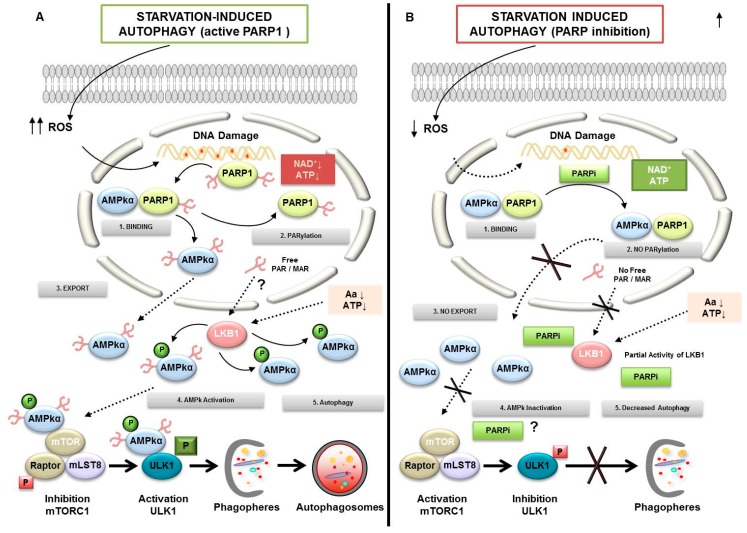
Regulation of autophagy by PARylation. (1) PARP1 forms a complex with AMPKα in nucleus. (**A**) PARP activation conditions: During starvation-induced autophagy, ROS release leads DNA damage and PARP1 overactivation. Self-PARylated PARP1 interact with AMPKα1 subunit (2). The complex is disrupted and PAR-AMPKα is exported to cytosol (3). The presence of PAR-AMPK and the continuous absence of amino acids and ATP depletion favor total activation of AMPKα population by LKB1, inhibition of mTORC1, interaction PAR-phospho-AMPK/ULK1, and autophagosome formation (4). LKB1 activity is presumably modified in a PARylation-dependent manner. (**B**) PARP inhibited: Starvation-induced ROS production was abrogated after PARP inhibition. Following AMPKα1/PARP1 interaction (1) the AMPKα1 subunit is not PARylated and AMPK nuclear export is inhibited (2 and 3). In spite of nutrient and energy depletion, AMPKα is inactive; mTORC1 is partially activated and interacts with ULK1, favoring its inhibition (4). Finally, the autophagosomes production will be delayed.

**Figure 9 cancers-12-00739-f009:**
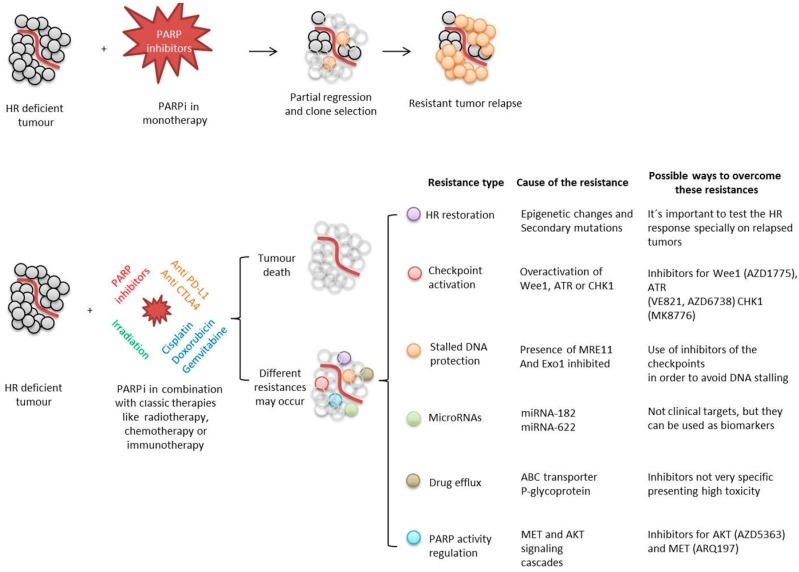
Summary of PARP inhibition approaches during cancer treatments, different possible outcomes and ways designed to overcome the possible appearance of resistances.

**Table 1 cancers-12-00739-t001:** Advanced clinical trials combining PARP inhibition with different classical therapies.

Status	Study Title	Conditions	Interventions	Phase	Number Enrolled	NCT Number
Active, not recruiting	A Phase 3 Randomized, Placebo-controlled Trial of Carboplatin and Paclitaxel With or Without Veliparib (ABT-888) in HER2-negative Metastatic or Locally Advanced Unresectable BRCA-associated Breast Cancer	Metastatic Breast Cancer	Drug: Paclitaxel Drug: Veliparib Drug: Carboplatin Other: Placebo	Phase3	513	NCT02163694
Active, not recruiting	Veliparib With Carboplatin and Paclitaxel and as Continuation Maintenance Therapy in Subjects With Newly Diagnosed Stage III or IV, High-grade Serous, Epithelial Ovarian, Fallopian Tube, or Primary Peritoneal Cancer	Ovarian Cancer Ovarian Neoplasm	Drug: Veliparib Drug: Paclitaxel Drug: Carboplatin Other: Placebo	Phase 3	1140	NCT02470585
Active, not recruiting (Has results)	Olaparib Treatment in BRCA Mutated Ovarian Cancer Patients After Complete or Partial Response to Platinum Chemotherapy	Platinum Sensitive BRCA Mutated Relapsed Ovarian Cancer Following Complete or Partial Response to Platinum Based Chemotherapy	Drug: Olaparib 300mg tablets Drug: Placebo to match olaparib 300mg	Phase 3	327	NCT01874353
Active, not recruiting	A Phase III Trial of Niraparib Versus Physician’s Choice in HER2 Negative, Germline BRCA Mutation-positive Breast Cancer Patients	Carcinoma of Breast Human Epidermal Growth Factor 2 Negative Carcinoma of Breast BRCA1 Gene Mutation BRCA2 Gene Mutation	Drug: niraparib Drug: Physician’s choice	Phase 3	306	NCT01905592
Active, not recruiting	A Study of Niraparib Maintenance Treatment in Patients With Advanced Ovarian Cancer Following Response on Front-Line Platinum-Based Chemotherapy	Ovarian Cancer	Drug: Niraparib Drug: Placebo	Phase 3	620	NCT02655016
Active, not recruiting (Has results)	A Study of Rucaparib as Switch Maintenance Following Platinum-Based Chemotherapy in Patients With Platinum-Sensitive, High-Grade Serous or Endometrioid Epithelial Ovarian, Primary Peritoneal or Fallopian Tube Cancer	Ovarian Cancer Fallopian Tube Cancer Peritoneal Cancer	Drug: Rucaparib Drug: Placebo	Phase 3	564	NCT01968213
Active, not recruiting (Has results)	Assessment of the Efficacy and Safety of Olaparib Monotherapy Versus Physicians Choice Chemotherapy in the Treatment of Metastatic Breast Cancer Patients With Germline BRCA1/2 Mutations.	Breast Cancer Metastatic BRCA 1 Gene Mutation BRCA 2 Gene Mutation	Drug: Olaparib Drug: Physician’s choice chemotherapy	Phase 3	302	NCT02000622
Active, not recruiting	Olaparib Maintenance Monotherapy in Patients With BRCA Mutated Ovarian Cancer Following First Line Platinum Based Chemotherapy.	Newly Diagnosed Advanced Ovarian Cancer FIGO Stage III-IV(and 4 more)	Drug: Olaparib 300mg tablets	Phase 3	451	NCT01844986
Active, not recruiting	Olaparib or Cediranib Maleate and Olaparib Compared With Standard Platinum-Based Chemotherapy in Treating Patients With Recurrent Platinum-Sensitive Ovarian, Fallopian Tube, or Primary Peritoneal Cancer	BRCA Rearrangement Deleterious BRCA1 Gene Mutation Deleterious BRCA2 Gene Mutation (and 13 more)	Drug: Carboplatin Drug: Cediranib Maleate Drug: Gemcitabine Hydrochloride (and 6 more)	Phase 3	549	NCT02446600

**Table 2 cancers-12-00739-t002:** Different clinical trials on cancer treatment combining PARP inhibition with anti-angiogenic strategies.

Status	Study Title	Conditions	Interventions	Phase	Number Enrolled	NCT Number
Recruiting	Phase 2, A Study of Niraparib Combined With Bevacizumab Maintenance Treatment in Patients With Advanced Ovarian Cancer Following Response on Front-Line Platinum-Based Chemotherapy	Ovarian Cancer Fallopian Tube Cancer Primary Peritoneal Carcinoma	Drug: Niraparib Biological: Bevacizumab	Phase 2	90	NCT03326193
Recruiting	A Study of Cediranib and Olaparib at Disease Worsening in Ovarian Cancer	Ovarian Cancer	Drug: Cediranib Drug: Olaparib	Not Applicable	30	NCT02681237
Recruiting	A Study of Fluzoparib Given in Combination With Apatinib in Ovarian or Breast Cancer Patients	Ovarian Cancer Triple Negative Breast Cancer	Drug: Fluzoparib Drug: Apatinib	Phase 1	76	NCT03075462
Recruiting	Phase 2 Multicohort Study to Evaluate the Safety and Efficacy of Novel Treatment Combinations in Patients With Recurrent Ovarian Cancer	Ovarian Cancer	Drug: Niraparib Drug: TSR-042 Drug: Bevacizumab	Phase 2	40	NCT03574779
Recruiting	Mesothelioma Stratified Therapy (MiST): A Multi-drug Phase II Trial in Malignant Mesothelioma	Mesothelioma, Malignant	Drug: Rucaparib Drug: Abemaciclib Drug: pembrolizumab & bemcentinib Drug: Atezolizumab & Bevacizumab	Phase 2	120	NCT03654833
Recruiting	Study Evaluating the Efficacy of Maintenance Olaparib and Cediranib or Olaparib Alone in Ovarian Cancer Patients	Ovarian Cancer	Drug: Olaparib Drug: Cediranib	Phase 3	618	NCT03278717
Completed	A Study of Cediranib and Olaparib at the Time Ovarian Cancer Worsens on Olaparib	Ovarian Cancer	Drug: Olaparib Drug: Cediranib	Phase 2	4	NCT02340611

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
