# Peer review of "The Multifactorial Role of PARP-1 in Tumor Microenvironment"

_cancers, 2020, doi:10.3390/cancers12030739_

Round 1

Reviewer 1 Report

I really enjoyed reading this review and learnt a lot.  A couple of tweaks and a little attention to the English (eg. lines 176-9) when editing are required but otherwise a credit to the journal.

Lines 242-6 need to include ovarian cancer here as this is the only cancer where a combination of chemotherapy with bevacizumab the VEGF inhibitor (continued as maintenance) has shown an overall survival benefit of 4-6 months (ICON 7, Oza 2013)

Lines 249-51 need to reference the recently reported PAOLA 1 study (ESMO 2019) which showed a PFS benefit for ovarian cancer patients treated with a combination of olaparib with bevacizumab versus bevacizumab alone - best in the BRCA mutated population but (interestingly and contradicting niraparib data), not seen in an HR proficient population (Ray-Coquard)

Lines 301-341 The hypothesis that mutational load in various cancers represents a balance between DNA damage and DNA repair mechanisms is somewhat lacking in evidence.  Copy-number cancers such as ovarian cancer and small cell lung cancer are not immunologically hot but have extraordinary levels of damaged DNA/chromosomes.....the neoantigenic hypothesis is mythical, the vast majority of mutations within expressed genes in cancers do NOT lead to the formation of neoantigens that are recognised by T cells (. C. Linnemann et al., Nat. Med. 21, 81–85 (2015). 17. Y. C. Lu et al., Clin. Cancer Res. 20, 3401–3410 (2014).) 

Author Response

Dear Reviewer,

We really thank you for your positive comments and we have introduced all the changes suggeted

Syncerely

F Javier Oliver, PhD

Reviewer 2 Report

The review by Martí et al. is a nice overview on the role of PARPs, mainly PARP1, in a variety of conditions characterizing the tumor microenvironment that may increase tumor aggressiveness and progression. The following changes are required before acceptance:

1) Since the review is mainly focused on PARP1, the title should be changed in: “The multifactorial role of PARP1 in tumor microenvironment”;

1) lines 178-179 “It has also been described how the PARP inhibitor veliparib can sensitize hypoxic cells in prostate and lungs to radiotherapy [54]: in the cited article, the PARP inhibitor was shown to increase radiosensitivity in both oxic and hypoxic cells, indicating that other mechanisms are involved besides a possible modulation of HIF1α expression (i.e. inhibition of the repair of DNA damage induced by ionizing radiations). This point should be clarified.

2) Figure 1: the caption of the Figure “Vasculogenic mimicry pathway and PARP in VM” is confusing taking into account that the authors have indicated in the text “VM” as the acronym of vasculogenic mimicry. Therefore the Figure caption should be corrected. The figure should be also modified: I would suggest to correct VM cells with “tumor cells”.

3) In regard to vasculogenic mimicry and VEGFR-1 the authors should cite and comment the following articles: Graziani G. et al. Antitumor activity of a novel anti-vascular endothelial growth factor receptor-1 monoclonal antibody that does not interfere with ligand binding. Oncotarget. 2016 Nov 8;7(45):72868-72885; Pagani E et al. Placenta growth factor and neuropilin-1 collaborate in promoting melanoma aggressiveness. Int J Oncol. 2016 Apr;48(4):1581-9; and Ruffini F et al. Cilengitide downmodulates invasiveness and vasculogenic mimicry of neuropilin 1 expressing melanoma cells through the inhibition of αvβ5 integrin. Int J Cancer. 2015 Mar 15;136(6):E545-58.

4) The name "Bevacizumab" (line 243) should be written with lower case (bevacizumab), since international standard drug names should not be capitalized.

5) In regard to the potential anti-angiogenic activity of PARP inhibitors, the following studies should be cited and commented: Tentori L et al. Poly(ADP-ribose) polymerase (PARP) inhibition or PARP-1 gene deletion reduces angiogenesis. Eur J Cancer. 2007 Sep;43(14):2124-33, and Lacal PM et al. Pharmacological inhibition of poly(ADP-ribose) polymerase activity down-regulates the expression of syndecan-4 and Id-1 in endothelial cells. Int J Oncol. 2009 Mar;34(3):861-72.

6) Figure 7: IFγ should be IFNγ.

7) The abbreviation MNNG (i.e., methylnitronitrosoguanidine) should be defined when first used.

8) Table 1. Advanced clinical trials combining PARP inhibition with different classical therapies: this table should indicate only clinical trials testing the PARP inhibitor in combination with other agents (as stated in the Table caption). For instance in the clinical trial NCT01945775 talazoparib is tested on its own; this observation applies also to NCT02184195 where olaparib is tested as maintenance therapy, etc. Therefore, the authors should revise the table including only clinical trials where PARP inhibitors are evaluated in combination with chemotherapeutic agents. Moreover, the title of the table should be changed in “Phase 3 clinical trials evaluating PARP inhibitors in combination with chemotherapy”.

9) Table 2. Different clinical trials on cancer treatment combining PARP inhibition with anti-angiogenic strategies: in the clinical trial NCT03168880 no PARP inhibitors nor anti-angiogenic agents are tested; actually, the NCT03168880 evaluates paclitaxel vs paclitaxel + carboplatin. Moreover, in the clinical trial NCT02734004 olaparib is combined with MEDI4736 (durvalumab) that is an anti-PD-L1 monoclonal antibody. Therefore, the authors should revise the table including only clinical trials where PARP inhibitors are evaluated in combination with anti-angiogenic agents.

10) A table including clinical trials testing PARP inhibitors in combination with immune checkpoint inhibitors should be added.

11) Figure 9: the acronym for PARP inhibitor “iPARP” should be corrected in “PARPi”.

12) In the "Conclusions and perspectives" section, when referring to additional synthetic lethal partners that might confer sensitivity to PARPi in wild-type BRCA tumors (lines 519-529), the following review should be cited: Faraoni I and Graziani G. Role of BRCA Mutations in Cancer Treatment with Poly(ADP-ribose) Polymerase (PARP) Inhibitors. Cancers (Basel). 2018 Dec 4;10(12):487.

13) The English language needs revision. Several typographical or grammatical errors are present throughout the manuscript; for instance:

-line 63: “zinq fingers” should be “zinc fingers”;

-line 90: “ester bound” should be “ester bond”;

-line 107: “referred as” should be “referred to as”

-line 116: “liberation” should be “release”;

-line 133: “phosphorilations” should be “phosphorylations”;

-line 135: “and the mediated” should be “and the one mediated”;

-lines 138-139: the sentence “Also remarkable is the ubiquitination at the 138 lysine 486 and 203 performed by the SUMO E3 ligase regulating PARP1 activity [38]” lacks the full stop.

-line 140: “like the described” should be corrected;

-line 180: “normal predominant HIF1α tissues” should be clarified;

-line 212: “by stimuli of VEGF by VEGFR2 activity and the co-receptor (NRP-1)” should be “by VEGF-mediated activation of VEGFR-2 and the co-receptor neuropilin-1 (NRP-1)”;

-line 313: “efficiency” should be “efficacy”;

-line 318: “a variety of stimulus” should be “a variety of stimuli”;

-line 351: “is present on all” should be “is present in all”;

-line 421: “leads to DNA damage and” should be “leads DNA damage and”.

Author Response

Dear Reviewer,

We really thank you for your positive comments and we have introduced all the changes suggeted. Concerning the new table you suggest to include clinical trials testing PARP inhibitors in combination with immune checkpoint inhibitors, we have only found clinical trials currently recruiting patients and in a very early stage.

Syncerely

F Javier Oliver, PhD

Reviewer 3 Report

This review by Martí JM et al, reports the available data on the interaction between PARPs, PAR and key features of tumor microenvironment including hypoxia, autophagy, tumor initiating cells, angiogenesis and cancer-associated immune response. The paper is straightforward, well written, and concise. Definitely deserves to be published and is a valuable contribution to the “cancers” journal. Some minor flaws need to be addressed before publication.

Minor points:

[1] Lines 128-135:

Functional aspects of PARP1 is an important issue, and could be extended in this paragraph. PARP1 regulates the c-Jun N-terminal kinase (JNK) pathway, which is a driver of tumor development and treatment response. Based on that, PARP inhibition could be potentially therapeutically beneficial in ovarian cancer taken the elevated JNK activity. Furthermore, PARP1 inhibitors promote Akt activity and mTOR signaling.

Relevant reference: Boussios S, et al. PARP Inhibitors in Ovarian Cancer: The Route to "Ithaca". Diagnostics (Basel). 2019 May 18;9(2). pii: E55.

[2] Lines 178-179:

Please, make here a comment about the mechanism of action of veliparib, which is the sensitizing effect to DNA-damaging treatments, such as chemotherapy and radiotherapy. Veliparib potentiates the effect of fractionated radiation through its impairment of both DNA single strand breaks and double strand breaks repair pathways.

Relevant reference: Boussios S, et al. Veliparib in ovarian cancer: a new synthetically lethal therapeutic approach. Invest New Drugs. 2020 Feb;38(1):181-193.

[3] Lines 242-246:

From the clinical perspective, beyond bevacizumab (PAOLA-1 and AVANOVA trials), there are available studies evaluating the antiangiogenic agent cediranib with the PARP inhibitor olaparib (NCT01116648, GY004, COCOS, OCTOVA, and CONCERTO trials), in several different settings. These studies should be added in table 2 as well.

Relevant reference: Boussios S, et al. Combined Strategies with Poly (ADP-Ribose) Polymerase (PARP) Inhibitors for the Treatment of Ovarian Cancer: A Literature Review. Diagnostics (Basel). 2019 Aug 1;9(3). pii: E87.

[4] Lines 263-266:

Please, clarify here that the production of pro-inflammatory cytokines, such as TNF, IL-6, INF, E-selectin, and ICAM-1 is the result of the interaction between PARP1 and the NF-κB pathway. PARP inhibition attenuates the upregulation of these factors in response to inflammatory stimuli. The loss of PARP1 activity inhibits proliferation and metastasis through anti-inflammatory mechanisms.

Relevant reference: Boussios S, et al. PARP Inhibitors in Ovarian Cancer: The Route to "Ithaca". Diagnostics (Basel). 2019 May 18;9(2). pii: E55.

[5] Lines 315-323:

Please, make here a comment specifically about the high-grade serous ovarian cancers. Tumors harboring BRCA1/2 mutations have demonstrated higher neoantigen burden, and CD3+ / CD8+ tumor-infiltrating lymphocytes. Therapeutically, increased levels of PD-1 and PD-L1 expression on tumor-infiltrating immune cells as compared to homologous recombination proficient tumors indicates that PD-1/PD-L1 inhibitors have a better efficacy in BRCA1/2-mutated high-grade serous ovarian cancer.

Relevant reference: Boussios S, et al. Combined Strategies with Poly (ADP-Ribose) Polymerase (PARP) Inhibitors for the Treatment of Ovarian Cancer: A Literature Review. Diagnostics (Basel). 2019 Aug 1;9(3). pii: E87.

Author Response

We thank this referee for for the effort in reviewing our manuscript and we have included the suggested references.

Reviewer 4 Report

In this review the authors debate the role of PARPs in tumor microenvironment. They start with a structural description of the proteins’ family and then they analyzed PARP involvement in different cancer condition: hypoxia, angiogenesis, cancer immune-microenvironment, autophagy and cancer initiating cells. The review is well conceived and written. Nevertheless, some minor revisions are required before the final publication.

  • pag 2, line 57: the group “other PARPs” should be not considered as a five point, because the authors discuss about 4 sub-families. This list should be misleading. It is sufficient delete the five point from the bullet list.

  • pag 3, line 63: Zinc inted of zinq

  • pag 3, line 66: this description should be more extensive and some references should be added: “ DNA-binding domain, formed by Zinc finger……….

  • The figures 4, 6, 8 and 9 should be supplied in high resolution, because they are difficult to read

  • pag 8, line 197: “Emerging…..progression.”, what studies? References are needed.

  • Lane 211: Figure 6: the caption title should be improved: "Vasculogenic mimicry pathway and PARP1" or "PARP1 in Vasculogenic mimicry". However, fig 6 could be postponed to line 232, because better contextualized.

  • Pag 9, line264:  TNFα, iNOS and IL-1β should be written in full.

  • Figure7, line 338: PD1 instead of PD11

  • Pag 12, lines358,361,362,363: ATM, MNR, TSC2 and ULK1 should be written in full.

Author Response

Dear Reviewer,

we thank you so much for your positive comments and your careful review. We have introduced all the suggested changes in the text and figures.

Sincerely

F. Javier Oliver, PhD